# Adventitious and Hairy Root Cultures of *Salvia apiana* as a Source of Rosmarinic Acid

**DOI:** 10.3390/ijms26073138

**Published:** 2025-03-28

**Authors:** Agata Krol, Adam Kokotkiewicz, Aleksandra Krolicka, Krzysztof Hinc, Anna Badura, Andzelika Lorenc, Urszula Marzec-Wroblewska, Adam Bucinski, Lukasz Kuzma, Maria Luczkiewicz

**Affiliations:** 1Department of Pharmacognosy, Faculty of Pharmacy, Medical University of Gdansk, Generała Józefa Hallera Street 107, 80-416 Gdansk, Poland; adam.kokotkiewicz@gumed.edu.pl (A.K.); mlucz@gumed.edu.pl (M.L.); 2Laboratory of Biologically Active Compounds, Intercollegiate Faculty of Biotechnology UG and MUG, University of Gdansk, Abrahama Street 58, 80-307 Gdańsk, Poland; aleksandra.krolicka@biotech.ug.edu.pl; 3Division of Molecular Bacteriology, Medical University of Gdansk, Dębinki Street 1, 80-211 Gdańsk, Poland; krzysztof.hinc@biotech.ug.edu.pl; 4Department of Biopharmacy, Faculty of Pharmacy, Collegium Medicum in Bydgoszcz, Nicolaus Copernicus University in Torun, Jagiellonska Street 15, 85-067 Bydgoszcz, Poland; anna.badura@cm.umk.pl (A.B.); andzelika.lorenc@cm.umk.pl (A.L.); u.marzec@cm.umk.pl (U.M.-W.); adam.bucinski@cm.umk.pl (A.B.); 5Department of Biology and Pharmaceutical Botany, Medical University of Lodz, 1 Muszyńskiego Street, 90-151 Lodz, Poland; lukasz.kuzma@umed.lodz.pl

**Keywords:** in vitro tissue culture, bioreactors, *Rhizobium rhizogenes*, white sage

## Abstract

For the first time, adventitious and hairy root cultures of *Salvia apiana* (white sage) have been established and analyzed for the content of secondary metabolites. Non-transformed roots derived from sterile seedlings were maintained on a full-strength IBA-supplemented SH medium. Adventitious roots yielded up to 44.5 mg/g and 18.7 mg/g DW rosmarinic acid when grown in shake flasks and immersion-column bioreactors, respectively. Transformed root cultures were established from *S. apiana* microshoots, infected with A4 and LBA9402 strains of *Rhizobium rhizogenes*. The obtained hairy root cultures (three and two clonal lines established using A4 and LBA9402 strains, respectively) were maintained in the PGR-free, full-strength SH medium. The most productive root line, established using A4 strain, accumulated rosmarinic acid at 38.1 and 39.6 mg/g DW when grown in shake flasks and spray bioreactors, respectively. Neither adventitious nor transformed roots of *S. apiana* produced diterpenoids, identified in roots of the field-grown plants, and instead proved to be a selective source of rosmarinic acid.

## 1. Introduction

Plants of the *Salvia* genus have a long-standing role in traditional medicine systems across the world [1,2,3], and several representatives of the genus have also found use in modern medicine. In particular, three species—*Salvia officinalis* L. (common sage), *Salvia sclarea* (clary sage), and *Salvia miltiorrhiza* Bunge (red sage)—are recognized for their therapeutic potential and have their respective monographs in the European and Chinese pharmacopeias [4,5]. The biological activity of sage plants is closely tied to their high content of terpenes (including mono-, di-, and triterpenes) and phenolics (including phenolic acids and flavonoids) [2,3,6]. However, the chemical composition of extracts and isolated fractions from the above plants can vary significantly depending on the harvesting site and phenophase, which directly impact the consistency of their biological activity in pharmaceutical applications [7].

One distinctive representative of *Salvia* genus is white sage (*S. apiana* Jeps., subg. *Audibertia*), with chemistry and biological properties that have been reviewed by Krol and co-workers [8]. The discussed plant, endemic to Southwestern United States and northwestern Mexico, uniquely combines the biosynthetic capabilities of various medicinal *Salvia* species [8]. The broad phytochemical profile of white sage includes monoterpene-rich volatile oil, nor- and abietane-type diterpenoids (including tanshinones and carnosic acid derivatives), C23-terpenoids (apiananes and hassananes), triterpenoids (oleanolic and ursolic acids), phenolic acids (including rosmarinic acid), and flavonoids of flavone (cirsimaritin) and flavanone (hesperidin) type [8]. The volatiles content in *S. apiana* is comparable to species which are considered the richest in this regard, like *S. officinalis* L. [5] or *Salvia fruticosa* Mill. [9], but notably, white sage is devoid of neurotoxic thujone [8]. Importantly, roots of *S. apiana* contain tanshinones, diterpenoid compounds typically restricted to a small subset of species within the genus and valuable because of their various biological activities [1,10,11]. The above characteristics make *S. apiana* particularly interesting from both practical and purely scientific standpoint. However, due to the endemic status of the plant, and overexploitation linked to increasing popular use of white sage [8], the availability of the raw material for phytochemical and biological research may soon become limited.

Numerous studies have validated the use of in vitro cultures of different *Salvia* species as an effective approach for obtaining phytocompounds of interest. So far, both cell and organ (adventitious roots and hairy roots, as well as the shoot cultures) systems have been developed as alternative sources of polyphenolic and terpenoid compounds, addressing the challenge of supply–demand imbalances for medicinal raw materials [12,13,14,15]. Importantly, studies on in vitro root cultures of *Salvia* spp. demonstrated that they are capable of producing a wide range of metabolites, ranging from simple phenolic acids to complex diterpenoids [12,15]. The literature also indicates that the phytochemical profile of transformed roots can differ significantly from non-transformed ones, as the transformation process itself may lead to the accumulation of compounds not present in non-transformed tissues [16]. However, research of this kind has so far not been conducted on *S. apiana*, except from the work of our research group, which has resulted in establishing microshoot cultures of white sage, capable of producing biologically active essential oil [17]. This validates further studies on this unusual plant, especially in the area of plant in vitro cultures.

The objective of the present research was to develop a novel in vitro system using adventitious and hairy roots of white sage. That system was designed as a new source of bioactive compounds, with a particular focus on phenolic compounds. Given the rich chemical composition of *S. apiana*, described earlier in the introduction, it was expected that the in vitro roots of the species might serve as a source of valuable polyphenols, such as rosmarinic and salvianolic acids, as well as diterpenoids, which are found in the field-grown plant [8]. These compounds are known as anti-tumor, antioxidant, and anti-inflammatory agents [11]. In this study, we present the establishment of two types of *S. apiana* root cultures (anatomical and hairy roots), the optimization of their growth conditions, and scaling-up experiments using different types of bioreactors. The obtained cultures were evaluated for growth parameters, and selected biomass samples were examined for secondary metabolite biosynthesis.

## 2. Results and Discussion

### 2.1. Establishment of In Vitro Root Cultures

The adventitious root culture of *S. apiana* was established through a multi-step process, involving seed sterilization, followed by germination of sterile plant material. The detailed protocol of the procedure was described in our previous study [17]. Subsequently, the root system was isolated from the seedling and transferred to the Schenk–Hildebrandt (SH) [18] medium containing 1.0 mg/L indole-3-butyric acid (IBA) and 30 g/L sucrose. After several 3-week intervals, the culture was stabilized. The biomass was characterized by rapid growth, manifested by the development of new, light brown roots, which tended to form spherical clusters.

The choice of medium type and its plant growth regulator (PGR) composition was based on both literature data and previous experimental paths with adventitious root cultures of plants belonging to Lamiaceae family. The full-strength SH medium, supplemented with 1.0 mg/L IBA, has previously been demonstrated to promote adventitious root growth in *Caryopteris* spp. [19] and *Scutellaria barbata* [20]. Moreover, IBA was shown to stimulate tanshinone production in non-transformed roots of *S. miltiorrhiza* [21]. Considering the above, and given that, in the current work, the IBA-supplemented, full-strength SH medium provided satisfactory growth of *S. apiana* adventitious roots, further experiments focused solely on optimizing the concentration of said auxin for subsequent lab-scale studies.

Hairy root cultures were initiated via the infection of leaf and stem explants, excised from *S. apiana* microshoots [17], with the A4, LBA9402, and ATCC15834 strains of *R. rhizogenes*. Twenty days post-infection, at the wound sites inoculated with strains LBA9402 and A4, small white root clusters with light brown callus at their base were observed (Figure 1A,B). The isolation of individual roots and their subculturing in the medium on the same composition, supplemented with antibiotics, allowed for the development of putative hairy root lines, obtained using A4 (three clones: TA1, TA2, and TA3) and LBA9402 strains (two root lines: TL1 and TL2). However, no hairy roots were observed after infection with the ATCC15834 strain. The subsequent subculturing of five hairy root lines at approximately 3-week intervals was performed using the SH_0_ medium. The roots, light brown in color, demonstrated stable growth, with no turbidity observed in the medium, indicating the absence of transforming bacteria in the culture environment. Our data are consistent with previous studies demonstrating that sage plants generally show high susceptibility to transformation by *R. rhizogenes*, especially strain LBA9402, which was particularly effective in inducing hairy roots in multiple *Salvia* species [22,23,24]. Previous reports also indicate that successful transformation with the ATCC15834 strain is possible [24,25,26,27]. However, in our study, this strain did not induce hairy root formation, which may be due to species-specific limitations. Further experiments are needed to elucidate these factors, as our research focused on developing transformed root cultures rather than evaluating the transformation efficiency of specific bacterial strains. This study also demonstrated that transformed roots of *S. apiana* can be effectively grown in the full-strength SH medium. This observation is in agreement with other reports which show that high-salt media provide a suitable environment for the growth of transformed root cultures of sage plants [24]. Based on available data, the growth medium was supplemented with 30 g/L sucrose. Previous studies demonstrated that such concentration of saccharose provides the optimum growth of transformed roots of several *Salvia* sp., including *S. miltiorrhiza* [24,28], *S. austriaca* [29], or *S. sclarea* [22].

To assess the genetic status of the established hairy root cultures and to verify the integration of *R. rhizogenes* T-DNA into the plant genome, PCR analysis was performed. For this purpose, two genes located in the TL-DNA (*rol*B and *rol*C) and two genes located in the TR-DNA (*mas2* and *TR*) of the Ri plasmid were used. The presence of all four genes (with TL-DNA and TR-DNA) was demonstrated for the clone TL1 obtained by transformation with strain LBA9402 (Figure A1). On the other hand, the clone TA2 obtained with strain A4 only showed the presence of TL-DNA (Figure A2). The TL-DNA, which contains the *rol* genes, is involved in hairy root formation, and the TR-DNA, which contains the *TR* and *mas2* genes, among others, is responsible for agropine synthesis and auxin biosynthesis [30]. It is noteworthy that only two reports in the scientific literature have previously described the incorporation of a TL-DNA and TR-DNA segment of the Ri plasmid into the genome of the *Salvia bulleyana* [31] and *S. miltiorrhiza* [32]. PCR analysis confirmed transgenic nature of obtained clonal lines. Furthermore, the absence of *vir*G genes in the biomass confirmed that the transforming bacteria were successfully eradicated from the culture using antibiotics.

### 2.2. Phytochemical Analysis

For the analysis of non-volatile phenolics, methanolic extracts from both in vivo (intact plant) and in vitro roots were prepared and analyzed using HPLC. The compounds of interest were phenolic acids, flavonoids, and diterpenoids, previously identified as major secondary metabolites of white sage [8]. The results of qualitative and quantitative LC-DAD-ESI-MS analysis are presented in Table 1 and Table 2, whereas representative chromatograms are included in the Appendix A (Figure A3). Noticeably, the extracts from roots of the intact plant were red, indicating the presence of tanshinones. The HPLC analysis of root samples revealed the presence of six diterpenoids, which were identified based on co-chromatography with standards (where available), as well as analysis of MS and DAD spectra and their comparison with literature data. The main metabolite of this group was cryptotanshinone (Figure A3; Table 1 and Table 2), which was accompanied by trace amounts of other diterpenes, including hydroxycarnosic acid, rosmanol, and tanshinone IIA (Figure A3, Table 1). Interestingly, the extract from roots of field-grown plant did not contain phenolic acids (including rosmarinic acid), despite the fact that these compounds had previously been identified in the underground parts of white sage [33]. On the other hand, the current work confirmed the presence of abietane-type diterpenoids, associated with biosynthetic pathway of carnosic acid, in roots of the intact plant (Figure A3), whereas, in previous studies, only tanshinone-type diterpenoids were identified in underground parts *S. apiana* [33]. Given the limited amount of data concerning the chemistry of white sage, particularly its underground parts, it is not possible to explain the reported differences. However, the available data on the accumulation of abietane diterpenes and rosmarinic acid distribution in other representatives of the genus, such as *S. rosmarinus* and *S. officinalis*, could be helpful in the interpretation of current results. Importantly, the biosynthesis of carnosic acid is associated with chloroplasts [25,34,35], and because of that, its presence in underground parts of sage plants is considered to be the result of it being transported from the aerial organs [34,35]. Studies on *S. rosmarinus* demonstrated that roots of the plant contain carnosic acid and carnosol, albeit in concentrations compared to rosemary leaves [35,36] and their contents also changed during the vegetative cycle [37]. In the other work, the presence of carnosic acid and carnosol was confirmed in leaves of common sage (*S. officinalis*), whereas roots of the plant contained only small amounts of the latter [34]. The distribution of rosmarinic acid in leaves and roots of the discussed plants followed similar pattern, with underground parts of sage plants always containing lower amounts of said metabolite compared to the aerial organs [34,35,36,38]. Studies on *S. miltiorrhiza* demonstrated that rosmarinic acid content in roots depend on harvest site [39], germplasm line and growth stage of the plant [40]. Given the above, differences in chemical profiles of white sage roots can stem from geographic location (plants grown in China [33] or Poland), vegetation stage, genetics, or a combination of factors.

**Table 1 ijms-26-03138-t001:** Characterization of compounds from roots of field-grown plant and in vitro cultures (EX0 experiment—Table 3) of *Salvia apiana* methanolic extracts determined by high-performance liquid chromatography coupled with diode array detection and electrospray ionization mass spectrometry (HPLC-DAD-ESI-MS).

Peak No. ^a^	Retention Time [min]	λ_max_ [nm]	Pseudomolecular ion (*m*/*z*)	Tentative Identification	Plant Sample ^b^	Ref.
1	17.16	323	[M − H]^−^: 385	Sinapic acid hexose	NT	[41,42]
2	25.93	229, 283, 331	[M − H]^−^: 359	Rosmarinic acid *	NT, TA1-3, TL1-2	[33,43,44]
3	29.21	328	[M − H]^−^: 373	Methyl rosmarinate	NT	[41]
4	32.63	338	[M − H]^−^: 313	Salvianolic acid F	NT	[41]
5	34.14	337	[M − H]^−^: 313	Salvianolic acid F isomer	NT	[41]
6	43.02	-	[M − H]^−^: 345	Rosmanol	FG	[33,45]
7	44.72	285	[M − H]^−^: 347	Hydroxycarnosic acid	FG	[44]
8	50.79	260	[M − H]^−^: 347	Hydroxycarnosic acid isomer	FG	[33]
9	58.66	222, 264	[M + H]^−^: 297	Cryptotanshinone *	FG	[33]
10	62.67	-	[M + H]^−^: 295	Tanshinone IIA *	FG	[33]
11	64.59	-	[M + H]^−^: 315	1,6,6,9a-tetramethyl-1,2,4,5,5a,6,7,8,9,9a-decahydrophenanthro[1,2-b]furan-10,11-dione	FG	[33]

^a^ The peak label corresponds to the numbering used in Figure A3. ^b^ Plant samples: in vitro cultures—adventitious roots (NT), transgenic root lines following infection with strains A4 (three clones: TA1-2) and LBA9402 (two root lines: TL1-2); field grown plant: roots (FG). * Identification of the compound using a reference standard.

In the current work, cryptotanshinone levels of white sage roots were quantified for the first time. Previous studies on white sage focused on the isolation and/or identification of the aforementioned constituent in the underground part of the plant [33,46]. The determined in *S. apiana* cryptotanshinone content was 3.71 mg/g DW, which is substantially higher compared to roots of *Salvia przewalskii* (2.0 mg/g), previously demonstrated by Wang and co-workers [47] to be the richest source of said compound among 58 sage species included in the study. Studies on *S. miltiorrhiza* show that cryptotanshinone levels in its roots may vary significantly depending on geographic location (0.9–3.6 mg/g) [48], as well as germplasm line and harvest date (0.27–5.36 mg/g) [39]. In consideration of the aforementioned findings, it can be posited that the roots of *S. apiana* serve as a substantial source of the discussed compound. However, further studies are necessary to determine cryptotanshinone levels in white sage specimens harvested from different geographical locations within its natural habitat, collected in different months/years to find possible intra- and interseasonal variations. If high contents of the discussed metabolite were confirmed, it would prove *S. apiana* to be a truly unique sage species, characterized by an exceptionally high concentration of volatile oil in leaves [8,17] and, at the same time, capable of accumulating substantial amounts of cryptotanshinone in the underground part of a plant.

Compared to field-grown roots of *S. apiana*, the metabolic profile of adventitious and hairy root cultures (EX0, Table 1) was substantially different. Both types of in vitro biomass accumulated only phenolic acids (Figure A3, Table 1), with rosmarinic acid being the main compound present in substantial amounts (Table 2). Neither adventitious nor transformed roots of white sage produced abietane-type diterpenes (carnosol, carnosic acid) or norditerpenes of tanshinone group. Literature data indicate that the capability of in vitro root cultures of sage plants (both transformed and non-transformed) to accumulate secondary metabolites characteristic for the respective intact plants vary significantly depending on the species studied and other factors. Their common characteristic, however, is the lack of abietane-type diterpenes, such as carnosic acid, because their biosynthesis takes place within chloroplasts [31,32,33].

The data concerning the specific chemistry of non-transformed roots of plants from Lamiaceae family are relatively scarce since the majority of studies focused on investigating hairy root cultures. For instance, the phytochemical profile of adventitious root culture of *Ocimum basilicum* was limited to rosmarinic acid and related phenolic compounds, including lithospermic acid and lithospermic acid B [49]. In the other work, the non-transformed root cultures of *S. miltiorrhiza*, a plant known for its high tanshinone content, demonstrated the ability to produce compounds of this type as well [21]. As far as hairy root cultures of *Salvia* spp. are concerned, reports indicate that infection with *R. rhizogenes* often alters the biosynthetic profile of the established biomass. For example, in the transgenic root culture of *Salvia corrugata*, established via infection with the A4 strain, diterpenoids agastol and ferruginol were identified, whereas non-transformed root cultures of the plant only accumulated agastol [23]. Studies on hairy roots of *S. miltiorrhiza* yielded mixed results, with the cultures either retaining the ability to produce tanshinones [21] or exhibiting a biosynthetic profile restricted to phenolic acids [31]. Another tanshinone-producing species with hairy roots that were incapable of producing these compounds is *S. bulleyana* [31]. The examples of in vitro cultures producing primarily or exclusively rosmarinic include transformed roots of *S. officinalis* [25], *S. virgata* [27], and *S. wagneriana* Pol. [50]. The phenomenon where hairy root cultures of certain *Salvia* species, like *S. bulleyana* and *S. apiana*, produce only phenolic acids (notably rosmarinic acid), while field-grown roots of the respective species also synthesize tanshinones, emphasizing the influence of growth environment on the regulation of metabolic pathways. Studies on *S. miltiorrhiza* have demonstrated that the production of tanshinones can by stimulated either by activating defense-related metabolic pathways via elicitation or genetic engineering by overexpressing key biosynthetic genes in the MEP pathway (such as *SmGGPPS* and *SmDXSII*) [51]. It was demonstrated that combining multiple elicitors—such as methyl jasmonate, yeast extract, and silver ions—can synergistically boost biosynthesis of tanshinones [52]. In fact, hairy root cultures of *S. miltiorrhiza*, exposed to 1 g/L yeast extract and 0.41 mM Ag^+^, accumulated the highest reported amounts of tanshinones to date, equal to 22 mg/g of DW [53]. Proteomic analysis included in the same report supported the role of stress defense and redox homeostasis in biosynthesis of secondary metabolites in hairy roots of *S. miltiorrhiza* [53]. Given the above, the absence of diterpenoids in adventitious and hairy roots of *S. apiana* roots may be related to restrictions of the in vitro environment (i.e., lack of certain environmental stressors, essential for biosynthesis of said compounds) and/or inherent properties of the in vitro biomass (i.e., lack of functional enzyme systems, present in the whole plant).

In order to fully evaluate the capability of the established root cultures to produce diterpenoids, the in vitro biomasses were subjected to hydrodistillation in a Clevenger-type apparatus. The rationale for the analysis was the results of the previous reports indicating that underground parts of selected sage species accumulate substantial amounts of volatile diterpenoids. Notably, the hydrodistillation of *S. leriifolia* Benth. [54] roots yielded essential oil rich in abietatriene (32.5–39.4%), labda-7,13-dien-15-ol acetate (23.2–30.8%), and ferruginol (7.9–12.0%). Given that the above compounds can be difficult to identify and quantify using HPLC, it was decided to isolate volatile fractions from transformed and non-transformed roots of white sage and determine their composition using GC-MS. However, the hydrodistillation of in vitro biomasses (adventitious roots and hairy roots lines TA1, TA2, TA3, TL1, and TL2) using Clevenger apparatus material did not reveal the presence of volumetrically measurable amounts of volatiles.

Quantitative analysis of in vitro biomasses extracts demonstrated that adventitious roots of white sage contain 17.22 mg/g DW rosmarinic acid. Transformation with *R. rhizogenes* did not enhance rosmarinic acid production, as its contents in hairy roots varied from 9.83 to 15.32 mg/g DW, depending on the clonal line (Table 3). Based on the determined levels of rosmarinic acid (ca. 1–1.7% DW), root cultures of *S. apiana* can be regarded as a moderately rich source of the discussed compound. The literature data indicate that rosmarinic acid content of in vitro biomass can vary significantly. Depending on the species and culture type, basal rosmarinic acid levels ranged from <1% to about 14%, whereas elicited biomass contained up to *ca* 16–25% of this metabolite [55,56]. While *S. apiana* root cultures demonstrate relatively low high rosmarinic acid yields compared to in vitro root cultures of other plants within the Lamiaceae family, the ultimate application potential of a culture system depends not only on rosmarinic acid content but also on factors such as the stability of plant biomass and the scalability of production [57,58]. The key advantages of transformed roots include their rapid growth and other favorable characteristics, though adventitious roots are also extensively investigated as a source of secondary metabolites, as they do not produce opine-like substrates, which could be toxic to mammals [15]. In consideration of these issues, both non-transformed and transgenic roots of *S. apiana* were selected for further shake flask and bioreactor experiments.

### 2.3. Shake Flask Cultures

The experiments involving adventitious root cultures of *S. apiana* were conducted in two stages. At first, the optimum concentration of IBA was selected. According to various reports, IBA is considered the most suitable auxin for inducing and proliferating adventitious root cultures of medicinal plants. It is typically used in concentrations ranging from 0.5 to 7.0 mg/L, though the optimal dosage needs to be experimentally determined for each species [59]. Given the above, the non-transformed roots of white sage were grown in liquid media with varying concentrations (0.5–5.0 mg/L) of said growth regulator and later evaluated for growth parameters (Figure A4). Most importantly, the medium containing 1.0–1.5 mg/L IBA promoted the fastest growth (for 1.0 mg/L IBA: 13.09 ± 1.68 g DW/L and Gi 1139.78 ± 173.42%; for 1.5 mg/L IBA: 14.37 ± 0.87 g DW/L and Gi 1373.51 ± 368.01%) and desired morphology of the cultivated biomass. IBA supplementation at a level below 1.0 mg/L resulted in low biomass yield and spontaneous callus formation, whereas concentrations above 2.0 mg/L did not significantly increase growth parameters and also triggered necrosis and callus formation in roots. Consequently, two IBA concentrations (1.0 and 1.5 mg/L) were selected for the second stage of the experiment, aimed at determining the growth kinetics of adventitious root culture. The experiment included different lighting regimes (photoperiod and darkness) and two different concentrations of IBA (1.0 and 1.5 mg/L). Additionally, the biomass was collected on the day corresponding to the highest Gi value and screened for rosmarinic acid content.

The rationale for conducting the experiment under photoperiod was the results of previous studies on *Salvia* spp. According to the literature, the production of carnosic acid and carnosol is associated with metabolic activity in chloroplasts, and thus light can potentially stimulate the production of these abietane-type diterpenoids [60]. In the current work, however, the adventitious root culture maintained under photoperiod grew exceptionally slow, and it did not exhibit a typical sigmoidal growth curve. Consequently, the experiment was stopped on day 45, with maximum Gi and DW values of ca. 300% and 4.59 g/L. Given the above, subsequent experiments on non-transformed root cultures were conducted in the absence of light. The growth curves plotted for adventitious roots maintained in darkness exhibited a typical sigmoidal pattern, though biomass cultured at a higher auxin concentration showed better growth dynamics (Figure 2). Adventitious roots grown in the SH medium supplemented with 1.5 mg/L IBA were characterized by a short lag phase, followed by exponential growth starting from the third day of the experiment and subsequent linear phase (days 21–33). The peak growth for the examined biomass was observed on day 45 of the experiment (Gi = 2782.69%, DW = 14.78 g/L). However, despite fast growth, after 40 days of the experiment, necrotic changes were noted in the root biomass, manifested as dark brown, mucilaginous clusters of root hairs within the culture core. Despite this, both culture variants (1.0 or 1.5 mg/L IBA) yielded comparable maximum DW values. For roots incubated in the medium supplemented with 1.5 mg/L IBA, DW reached 15.74 g/L on day 33, while at a lower auxin concentration, the maximum DW concentration was 16.50 g/L on day 48. HPLC-DAD-ESI-MS analyses revealed that the tested biomasses selectively accumulated rosmarinic acid at levels of 44.49 and 31.70 mg/g DW on days 33 and 45 of the experiment, respectively (Figure 2). The shape of the growth curves suggested that the optimal cultivation period in bioreactor should not exceed 48 days.

As mentioned earlier, reports on secondary metabolite production in adventitious root cultures of plants from mint families are relatively scarce, and thus, the amount of data for comparative purposes is limited. Moreover, detailed growth profiles of the investigated cultures were plotted only in selected studies on this topic. For instance, the growth profile of *O. basilicum* adventitious root culture was similar to non-transformed roots of *S. apiana* established in the current work: the maximum DW content (ca. 13 g/L) was reached after 7 weeks, and the corresponding rosmarinic acid concentration was 45 mg/g DW [49]. In the other studies, the cultivation time of roots maintained in the liquid medium was fixed. Adventitious root culture of *S. fruticosa*, for example, reached ca. 5.5 g/L DW content over 4-week period, with rosmarinic acid content equal to ca. 25 mg/g [61]. Noticeably higher values were recorded for non-transformed root cultures of *Origanum dictamnus*. After 40-day cultivation in the liquid medium, the investigated adventitious root lines reached a DW content of up to 11–18 g/L and a rosmarinic acid concentration of 29–66 mg/g DW [62].

The growth curves of five hairy root lines of *S. apiana* are presented in Figure 3. Hairy roots of lines TA1, TA2, and TA3 entered the stationary phase on about the 50th day of cultivation. The highest biomass yield was recorded between days 50 and 60, depending on the clone, with maximum Gi values ranging from 1904.03% to 2610.33% and DW from 14.60 to 16.54 g/L. However, after day 50, the share of necrotic roots gradually increased, signaling the onset of the death phase. A similar trend was observed for root lines TL1 and TL2, with the highest amount of biomass also recorded between days 50 and 60 (maximum Gi values of 1916.98–2430.78% and DW of 15.78–16.28 g/L). Observed differences for biomass production between various hairy root lines could result from several factors, including the level of gene expression, the chromosomal location of the Ri T-DNA insertion, its length, and the number of copies present in the plant genome [26,63]. It is worth noting that, in comparison to non-transformed biomass, transformed roots of white sage showed somehow less dynamic growth. This phenomenon is rather unusual, as transgenic organs are generally considered to exhibit a higher growth rate compared to non-transformed tissues [16]. These results also contrast with those described in previous research on *Salvia* species, which documented faster progression through the respective growth phases, and an overall growth cycle not exceeding 6 weeks [23,31,32,64,65]. For example, *S. nemorosa* hairy roots reached the stationary phase after two weeks of cultivation; however, this was associated with approximately two times lower dry weight (7.1 g DW/L) [26] compared to the maximum dry weight of *S. apiana* roots (TA1 clone) on day 50 (15.9 g DW/L). In turn, an analysis of rosmarinic acid content reveals that previous studies on its production in transformed root cultures of *Salvia* species have shown its levels peak during the stationary phase [26,32]. For clone TA1, the rosmarinic acid content on day 50 was 38.14 mg/g DW, a value comparable to maximum concentration of this metabolite in hairy roots of *S. bulleyana*, obtained using A4 strain, equal to 39.6 mg/g DW [31].

The growth characteristics of *S. apiana* hairy root lines, along with phytochemical analyses, provided the basis for selecting culture for bioreactor experiments. For this purpose, clone TA1 was selected due to its relatively high rosmarinic acid content, favorable morphological characteristics, and dynamic growth curve with a high maximum Gi value, recorded on day 50 (2264.58%). Based on the growth profile, we decided to run the experiment for 60 days.

### 2.4. Bioreactor Experiments

The design of reactor vessels used in the experiment was tailored to meet the requirements of the root cultures and provide maximum biomass productivity. The installations with a working volume of 600 mL, employed in the current study, included the modified air-lift bioreactor (ICB) and a nutrient mist bioreactor (SB) (Figure A5). The above-mentioned systems are commonly employed to scale-up in vitro root cultures of medicinal plants [66]. Both adventitious and hairy (TA1 clone) root cultures were used in the experiments, according to the details given in Table 3.

The roots grown in the SB (both transformed and non-transformed ones) remained viable and were light brown in color after the cultivation period (Figure 1D,E). The quality of roots grown in the ICB system was reduced, which was noticeable, especially in the case of non-transformed culture, and manifested with the presence of dark brown, rigid, and necrotic biomass at the center of the basket (Figure 1C). Additionally, small, round callus fragments, distributed throughout adventitious root biomass, were also observed in the ICB. The highest Gi (1453.37%) and DW (14.51 g/L) rates were recorded for hairy roots cultured in the SB system. The reported values were approximately 3-fold higher than those of transgenic biomass, grown in the ICB, and 2- to 3-fold higher than those registered for adventitious roots grown in both systems (Figure 4).

The loss of viability of bioreactor-grown root cultures is a common challenge in biomass production at higher working volumes. Similar issues have been reported during bioreactor experiments involving *S. corrugata* [23] and *S. austriaca* [67]. The limitation of biomass growth is a well-documented phenomenon in in vitro systems. Biomasses with high physiological demands, such as roots, and particularly susceptible to restricted oxygen transfer and nutrient availability [66]. In our study, a design modification of the SB was implemented by utilizing multiple smaller inoculation sites, which significantly improved the quality of the harvested biomass. The developed system demonstrated greater compatibility with the physiological and morphological requirements of *S. apiana* transformed roots, providing better conditions for root proliferation compared to the modified air-lift system used in our research. Similar observations were previously made by Jaremicz and co-workers [68] who compared growth rates of *Hyoscyamus niger* hairy roots in bubble column and spray bioreactors. As in the current work, the spray bioreactor provided higher DW yield thanks uniform distribution of inocular biomass in the reaction vessel. The non-transformed roots of white sage, however, showed comparable growth rates in both ICB and SB systems (Figure 4).

Phytochemical analysis revealed that bioreactor cultures of both transformed and non-transformed roots selectively produced rosmarinic acid. The highest content of this metabolite (39.62 mg/g DW) and productivity (9.66 mg/L·day) were recorded in hairy root cultures grown in the SB system (Figure 4). In contrast, adventitious root cultures produced significantly lower amounts of rosmarinic acid, with concentrations ranging from 11.46 to 18.70 mg/g DW and productivities between 1.44 and 3.58 mg/L·day, depending on the bioreactor type and IBA concentration. This aligns with observations from previous studies. For instance, in the experiments involving hairy roots of *S. officinalis*, grown in sprinkle-type bioreactors, comparable results were achieved, with reported contents of rosmarinic acid equal to 34.65 mg/g DW [69].

The results of bioreactor experiments involving in vitro root cultures of *S. apiana* indicate that both types of biomass (transformed and non-transformed ones) are suitable for further studies using different bioreactor types. As mentioned earlier, hairy roots of white sage grown in the SB system proved to be superior in terms of growth rate and rosmarinic acid content. However, scaling-up of gas-phase bioreactors faces certain challenges. Importantly, the effective use of space within the growth vessel requires uniform distribution of the inoculum, which becomes a problem when dealing with larger bioreactors. This issue can be addressed by employing a hybrid system which initially operates as a column bioreactor, in order to immobilize roots onto the internal hooks/loops placed inside the vessel. The medium is subsequently drained and applied in the form of mist till the end of an experiment. An example of system of this type is a 500 L hybrid bioreactor used to grow hairy roots of *Datura stramonium* [70] which, however, was merely a prototype installation. Nutrient mist bioreactors with a rotatable culture bed, reaching a scale of 1000 L, have also been developed [71]. So far, the only attempt to commercially grow hairy roots in a gas-phase bioreactor was by the Swiss company ROOTec, which is no longer in operation [71]. The performance of gas-phase bioreactors can be improved by employing nutrient dispersion systems capable of producing fine mist, which facilitates gas and nutrient exchange and reduces mechanical stress. However, installations of this type are still more material-consuming and difficult to operate at a larger scale, compared to simpler bubble-column bioreactors BCB). Immersion systems such as BCB or balloon bioreactor, on the other hand, can be successfully used to grow non-transformed roots. Unlike gas-phase systems, bioreactors of this type are easier to scale up and maintain. For instance, 10,000 L installations have been developed to produce non-transformed roots of *Panax ginseng* for commercial purposes [72]. Although bioreactor-grown adventitious roots of *S. apiana* showed slower growth and lower rosmarinic acid content compared to hairy roots (Figure 4), the advantage of this type of biomass is stability, as well as lack of bacterial DNA and opine substrates [72]. Because of that, non-transformed roots are perceived as more natural and safe and may thus serve as an alternative source of plant biomass and/or specific secondary metabolites.

## 3. Materials and Methods

### 3.1. Reagents and General Procedures

All reagents used for plant in vitro culture experiments were supplied by Sigma-Aldrich (St. Louis, MO, USA). Ultrapure water for in vitro experiments and phytochemical analysis was obtained with the Elix/Synergy system (Merck KGaA, Darmstadt, Germany). Solvents used for extract preparation were from POCH (Gliwice, Poland), and reagents used for HPLC analysis were from Merck KGaA. Standard compounds used in phytochemical analysis were from Sigma-Aldrich.

The root cultures were maintained in the growth chamber at the stable temperature of 24 ± 2 °C. Depending on the cell line, the in vitro root cultures were cultivated either under a photoperiod (16 h/8 h light/dark; white fluorescent light with the intensity 88.8 μmol/m^2^s, Philips TLD 35 W/33 tubes, Amsterdam, The Netherlands) or/and in complete darkness.

### 3.2. Plant Material

The use of plant material in the study complies with relevant institutional, national, and international guidelines and legislation. Roots of field-grown *S. apiana* (employed in phytochemical analysis) were harvested from specimens grown in field at horticultural company Ogrody Ziołowe, located in Bielany Wrocławskie (Poland). Fresh roots of field-grown white sage were dried in a forced convection oven (FD 115, Binder, Tuttingen, Germany) at 30 °C for 48 h. The identity of the plant species was confirmed by Marcin Gorniak and Aleksandra M. Naczk, according to the previously described protocol [17]. The voucher specimen was deposited in the Herbarium of the Medical Plant Garden at the Department of Pharmacognosy, Medical University of Gdansk, Poland (the sample number S.a. 04/2023).

Sterile white sage seedlings, used to establish adventitious root line of *S. apiana*, were germinated from authenticated seeds (harvested in the year 2018; Strictly Medicinal Seeds, Williams, AZ, USA), according to the protocol detailed in our previous study [17].

*S. apiana* microshoots, used to establish hairy root cultures, were obtained previously from the top part of the hypocotyl with cotyledons of sterile white sage seedlings. The shoots were grown on Schenk–Hildebrandt (SH) medium supplemented with 0.22 mg/L thidiazuron (TDZ) and 2.0 mg/L 6-benzylaminopurine (BAP) [17] and subcultured at 4-week intervals. The culture medium was solidified with agar (0.6% *w*/*v*).

### 3.3. Establishment of Adventitious Root Culture

For adventitious root culture initiation, the root section of single, aseptically germinated *S. apiana* seedling [17] was excised and transferred into liquid Schenk–Hildebrandt (SH) medium [18] supplemented with indole-3-butyric acid (IBA) at 1 mg/L and 3.0% (*w*/*v*) sucrose. The initial adventitious root culture was maintained in the dark, under agitation at 120 rpm (INNOVA 2300, Brunswick Scientific, Edison, NJ, USA; φ25.4 mm), and subcultured at 3-week intervals for 6 months, using the SH medium with the same composition. As a result, a stable adventitious root culture of *S. apiana* was established. The obtained biomass was evaluated for polyphenol and terpenoid content (EX0, Table 3), and it also served as a source of inoculum for further in vitro experiments.

### 3.4. Establishment of Hairy Root Cultures and Confirmation of Transformation

In order to induce hairy roots, *S. apiana* microshoots [17] were infected with *Rhizobium rhizogenes* strains: ATCC A4, 15834, and LBA9402, cultivated for 48 h on yeast extract beef (YEB) agar medium with 200 μM of acetosyringone at 26 °C in the dark. Inoculation was performed on the leaf and stem sections by wounding with a sterile needle, dipped in bacterial culture grown on the solid medium. Following inoculation, *R. rhizogenes* and microshoots were co-cultivated in the dark for three days on solid (0.6% *w*/*v* agar) phytohormone-free SH medium, supplemented with 3.0% (*w*/*v*) sucrose (SH_0_). Subsequently, the explants were placed onto the solid SH_0_ medium supplemented with antibiotics: cefotaxime sodium at 500 mg/L (Duchefa Biochemie, Haarlem, The Netherlands) and carbenicillin disodium at 500 mg/L (Carl Roth, Karlsruhe, Germany) and kept in the dark. After 4 weeks, three individual root lines were isolated from explants infected with *R. rhizogenes* strain A4 (clones TA1, TA2, and TA3) and two from strain LBA9402 (clones TL1 and TL2). The obtained hairy root clones were transferred into the liquid SH_0_ medium supplemented with the aforementioned antibiotics, and maintained in the dark, under agitation (120 rpm). Over a period of 3 months, the respective clonal lines were sub-cultured six times using the same medium (approximately 2 weeks per passage) and eventually moved to antibiotics-free medium. The established hairy root cultures were subscultured at 3-week intervals and evaluated for secondary metabolite content (EX0, Table 1). The biomasses served as a source of inoculum for further in vitro experiments.

The transgenic nature of the established root lines was confirmed in the analysis of two clonal lines: TA2 and TL1. DNA from the aforementioned root lines, as well as untransformed roots (negative control), was extracted from 100 mg of plant tissue using CTAB method by Bekesiova et al. [73]. Primers for gene amplification of the TL-DNA and TR-DNA regions were designed based on the DNA sequence of the pRi plasmid from *R. rhizogenes* (GenBank accession number CP044124.1). Clone Manager 9 Professional Edition and Oligo Primer Analysis Software version 7.60 were used to design the primers. The annealing temperature for the primer pairs was between 53.5 and 67 °C and the amplicon length ranged from 273 to 1672 bp. The oligonucleotide primers for gene detection were: *rol*B (Forward 5′ GCTCTTGCAGTGCTAGATTT 3′; Reverse 5′ GAAGGTGCAAGCTACCTCTC 3′) and *rol*C (F 5′ CTCCTGACATCAAACTCGTC 3′; R 5′ TGCTTCGAGTTATGGGTACA 3′) genes for TL fragment of the Ri plasmid, *mas2* (F 5′ GCGCATCCCGAGGCGATG; R 5′ AGGTCTGGCGATCGCGAGGA) and *ags* (F 5′ CGGAAATTGTGGCTCGTTGTGGAC; R 5′ AATCGTTCAGAGAGCGTCCGAAGTT) genes for TR fragment of the Ri plasmid and *vir*G (F 5′ ACTGAATATCAGGCAACGCC 3′; R 5′ GCGTCAAAGAAATAGCCAGC 3′) gene outside of the T-DNA fragment.

### 3.5. Shake Flask Cultures

Studies on agitated adventitious and hairy root cultures of white sage included experiments EX1, EX2 and EX4, as presented in the details in Table 3. The collected root samples were assessed for growth parameters (fresh weight—FW, dry weight—DW, growth index—Gi; Section 3.7), morphological features, and secondary metabolite content (selected samples, Table 3).

### 3.6. Bioreactor Experiments

Adventitious (NT) and transgenic (TA) root lines were cultivated in the immersion-column bioreactor (ICB), equipped with stainless steel basket for root immobilization and a barbotage-based mixing and aeration system, as well as in the custom-made spray bioreactor (SB). The design and specification of the ICB were described in earlier works [68,74]. The SB system used was based on previous study [74], adapted for the purpose of current work by placing a stainless-steel scaffolding, for root immobilization, inside the growth vessel. It consisted of 6 hexagonally arranged vertical rods, fastened rigidly to a perforated plate located above medium surface. The inoculum was uniformly distributed in 9 mesh baskets (20 × 20 × 20 mm), attached to the rods in three layers, in an alternating fashion. The schematic drawings of both systems are presented in Figure A5.

The details of experiments involving bioreactor-grown adventitious (EX3) and hairy roots (EX5) of *S. apiana* are presented in Table 3. The harvested root samples were assessed for growth rate (FW, Gi, DW, Section 3), morphological features, and secondary metabolite content (Table 3).

### 3.7. Determination of Growth Parameters

After removing the liquid medium and washing the roots with distilled water, the plant material’s fresh weight (FW) was measured. Gi was determined using the following formula:Gi = (FW_x_ − FW_0_)/FW_0_ × 100
where Gi represents the growth index, FW_0_ is the initial fresh weight of the inoculum, and FW_x_ is the fresh weight after X days of cultivation. For dry weight (DW) determination, the biomass was frozen for at least 24 h at −10 °C, and subsequently freeze-dried (1 × 10^−1^ mbar, 72 h, Steris Lyovac GT2 lyophilizer; Finn-Aqua Santasalo-Sohlberg, Tuusula, Finland).

### 3.8. Extraction Procedures

For the analysis of non-volatile constituents, 5.0 g samples (dried field-grown roots, freeze-dried adventitious and hairy roots) were crushed in a mortar and extracted with methanol using a magnetic stirrer (3 × 200 mL, 2 × 2.0 h + 1 × 12.0 h; 21 °C, 200 rpm). After filtration, the extracts were combined and vacuum-concentrated (30 °C) using a Rotavapor R-300 (BÜCHI, Switzerland) to a final volume of 100 mL. Prior to HPLC-DAD-ESI-MS analysis, the extracts were centrifuged for 10 min at 7500× *g*. For the materials available in smaller amounts, the sample size and final extract volume were 1 g and 20 mL, respectively.

For essential oil isolation, 20 g samples of freeze-dried adventitious and hairy roots, were subjected to hydrodistillation using Clevenger apparatus according to European pharmacopeia (400 mL of distilled water, 3 h). The content of volatiles was determined volumetrically.

### 3.9. Qualitative and Quantitative HPLC-DAD-ESI-MS Analysis

Qualitative and quantitative HPLC analyses of non-volatile polyphenols and terpenes were conducted using LC-DAD-ESI-MS system (Shimadzu, Japan), which included two solvent pumps (LC-20AD), an autosampler (SIL-20AC, 15 °C), a diode array detector (SPD-M20A), a mass spectrometry detector (ESI 2010EV), a column oven (CTO-20AC, 30 °C), and a degasser (DGU-20A3). The separation was conducted on a C18 column (Phenomenex Luna, 5 µm, 250 × 4.6 mm), using a mobile phase consisting of acetonitrile (B) and 0.1% acetic acid (A) at a flow rate of 0.8 mL/min. The elution was carried out in a gradient system according to the program: 0 min: 0% B, 0–70 min—linear change/increasing 0–100% B, 90–100 min: 100% B, 100 min—0% B, 105 min—0% B. After chromatographic analysis, the column was equilibrated for five minutes to the initial composition of the mobile phase, namely 0%B. Mass spectrometric detection of the compounds was performed in the positive and negative ion mode over the range *m*/*z* 100–800. The analytes were identified by comparing their retention times and UV/MS spectra with those of the standards (Sigma-Aldrich). Rosmarinic acid (RA) and cryptotanshinone (CT) were quantified based on peak area (280 nm) using the respective calibration curves. The presented values represent the means of at least three replicates.

For RA measurements, the method was validated by assessing linearity (5.7 × 10^−4^ − 2.33 mg/mL, regression equation y = 1,996,951 × −10,789, r = 1.000), intra-day precision (%RSD, *n* = 6, 0.44), inter-day precision (%RSD, *n* = 6, 1.43), LOD (S/N = 3, *n* = 3, 2.8 × 10^−4^ mg/mL), LOQ (S/N = 10, *n* = 3, 5.7 × 10^−4^ mg/mL) and accuracy (% recovery at 3 levels, *n* = 9 (3 + 3 + 3), 100.57%, %RSD 0.9). For CT determination, linearity was assessed (regression equation y = 20,000,000 × −620549, r = 0.9954.

### 3.10. Statistical Analysis

Statistical analysis was performed using Statistica PQStat Software version 1.8.6 (Poznań, Poland). The Lilliefors test was employed to verify the hypothesis of non-significance of the difference in the distribution of the study variable with a normal distribution. Leven’s test was applied to evaluate the equality of variances. Group comparisons were made using analysis of variance followed by the LSD test, with a significance level set at *p* = 0.05.

## 4. Conclusions

The presented work is the first report on in vitro systems of *S. apiana*, based on transformed and non-transformed roots of white sage. Both mentioned types of biomass are regarded as a scalable and sustainable platform for the production of high-value secondary metabolites for pharmaceutical and nutraceutical industries [15,16]. The cultures established in the presented study can be considered a promising in vitro platform for the production of rosmarinic acid which exhibits a number of biological activities, including antioxidant, anti-inflammatory, and even anti-tumor effects [75]. Besides the clinically relevant aspects of rosmarinic acid, this compound is also valuable due to its potential as a food preservative and cosmetic ingredient. RA-rich rosemary extracts are used as food additive, exhibiting both antioxidant and antimicrobial effects, and thus extending shelf life of products. In this regard, they are often preferable to synthetic antioxidants due to raising concerns regarding the safety of compounds such as BHT, BHA, and TBHQ. Rosmarinic acid is also investigated as a pigment-stabilizing agent which could be used to preserve color of beverages containing carotenes and anthocyanins. For large-scale production, biotechnological approaches such as plant cell cultures, elicitor treatments, as well as bioreactor cultivation are employed to enhance rosmarinic acid yield, while advanced, environmentally friendly downstream processing techniques like ionic liquid-based extraction and ion-exchange centrifugal partition chromatography can ensure high purity and sustainability [58]. The experiments demonstrated that transgenic root cultures of white sage, grown in spray bioreactors, provided the highest rosmarinic acid content (39.62 mg/g DW) and productivity (9.66 mg/L∙per day) of white sage. The study also revealed distinct differences in the metabolic profiles of in vitro and field-grown roots of *S. apiana*. The underground parts of wild-grown plants were shown to produce cryptotanshinone, whereas the in vitro cultures selectively accumulated rosmarinic acid. Further studies will be aimed at improving the productivity of the developed in vitro systems with respect to rosmarinic acid. For this purpose, growth media of different strength (1/2SH or 1/3 SH) and elicitors of different types will be used. Elicitation strategies will also be employed to trigger the biosynthesis of tanshinone constituents in transformed and non-transformed root cultures of white sage. As far as bioreactor cultures of *S. apiana* are concerned, the experiments will be focused on improving medium delivery systems in gas-phase installations, as well as optimizing culture conditions in airlift bioreactors. Aside from in vitro culture experiments, the results of the work also encourage further studies on wild-grown roots of *S. apiana*, which can be considered as a potential source of cryptotanshinone.

## Figures and Tables

**Figure 1 ijms-26-03138-f001:**
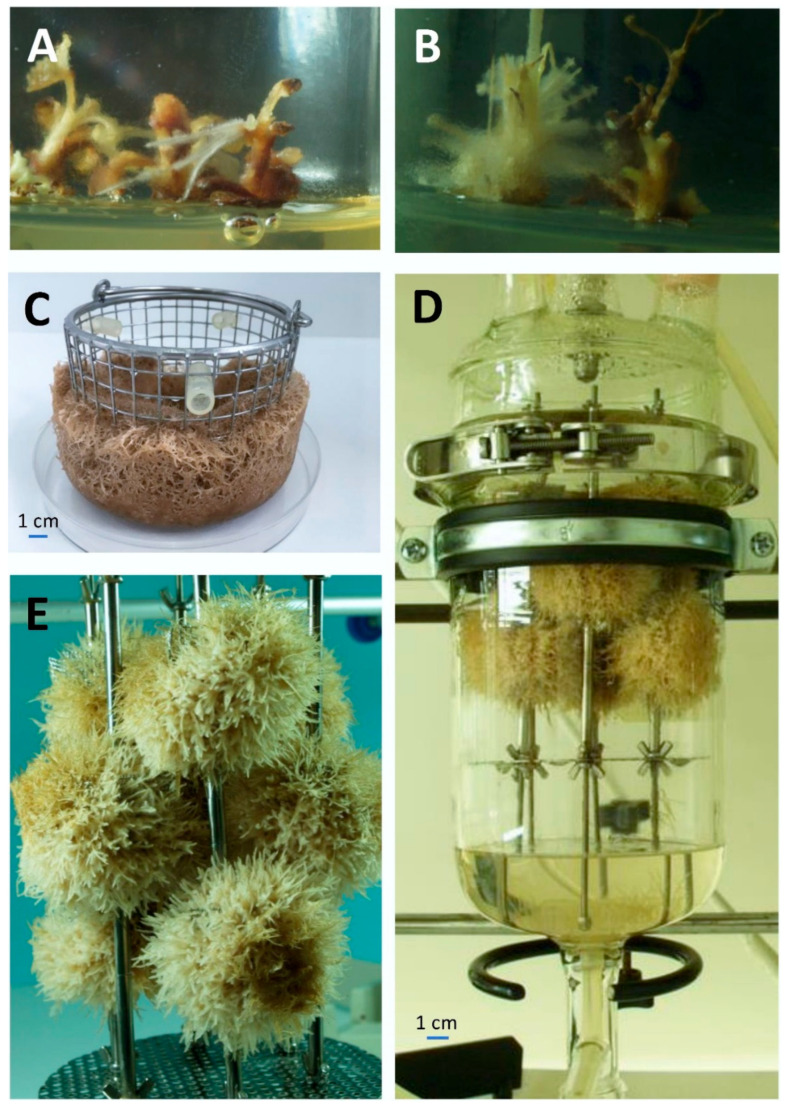
*Salvia apiana* cultures in the studied in vitro systems: (**A**) stem explants of *S. apiana* microshoots infected with *R. rhizogenes* strain LBA 9402 with hairy roots, cultivated for three weeks on solid SH medium, supplemented with antibiotics; (**B**) stem explants of *S. apiana* microshoots infected with *R. rhizogenes* strain A4 with hairy roots, cultivated for three weeks on solid SH medium, supplemented with antibiotics; (**C**) biomass of adventitious roots of *S. apiana* immobilized in bioreactor basket, grown in immersion-column bioreactor for 40 days in SH medium supplemented with 1.0 mg/L IBA; (**D**) biomass of transgenic roots clone TA2 (A4) of *S. apiana* in SG bioreactor grown for 60 days in SH0 medium; (**E**) biomass of adventitious roots of *S. apiana* immobilized in bioreactor baskets, grown in spray bioreactor for 48 days in SH medium supplemented with 1.5 mg/L IBA.

**Figure 2 ijms-26-03138-f002:**
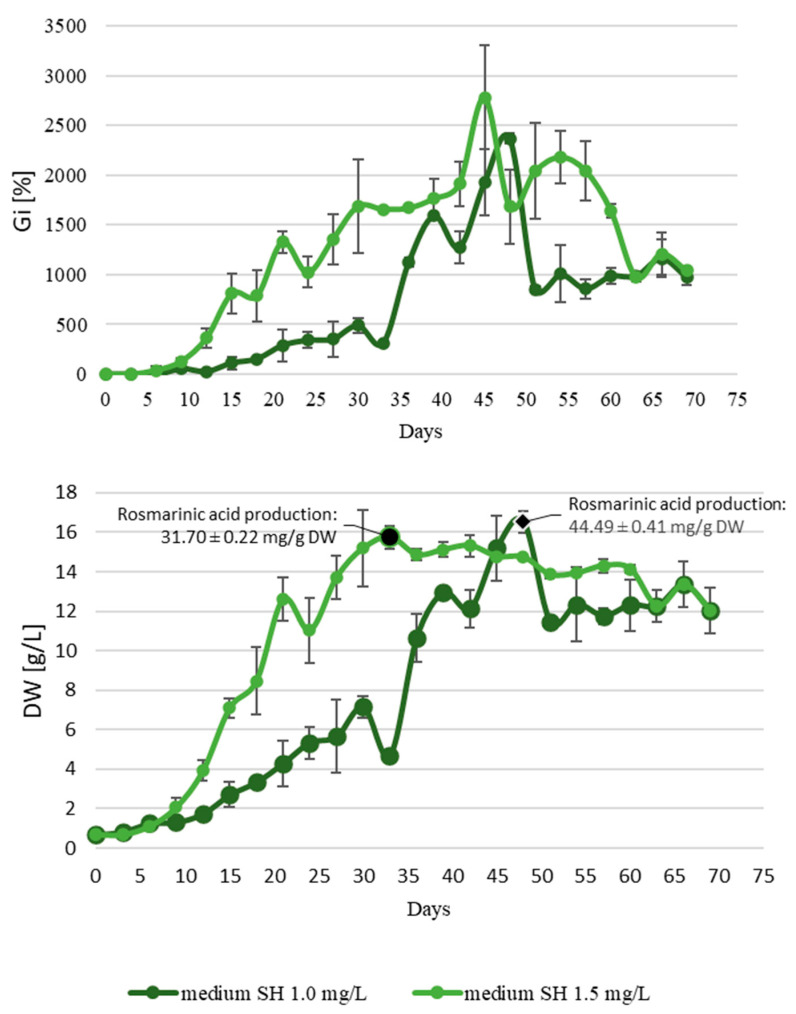
Growth and rosmarinic acid content in adventitious root culture of *S. apiana*, cultivated in SH medium with 1.0 or 1.5 mg/L IBA in the dark. Measurements were conducted over a 69-day growth period, with samples collected at 3-day intervals. Data are presented as the mean of triplicate experiments, with bars representing standard errors.

**Figure 3 ijms-26-03138-f003:**
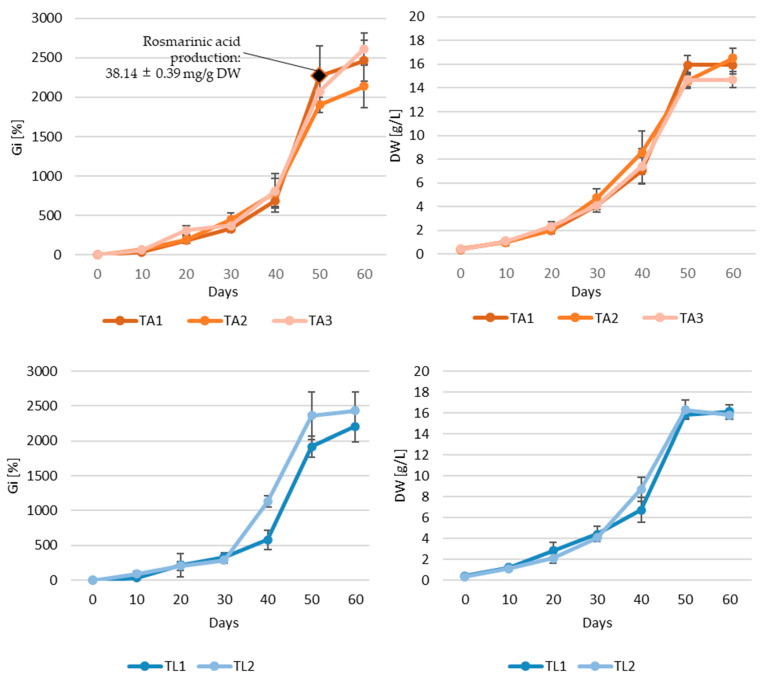
Growth and rosmarinic acid content in transgenic root cultures of *S. apiana* (TA1, TA2, TA3, TL1, and TL2), cultivated in SH_0_ medium in the dark. Measurements were conducted over a 60-day growth period, with samples collected at 10-day intervals. Data are presented as the mean of six replicate experiments, with bars representing standard errors.

**Figure 4 ijms-26-03138-f004:**
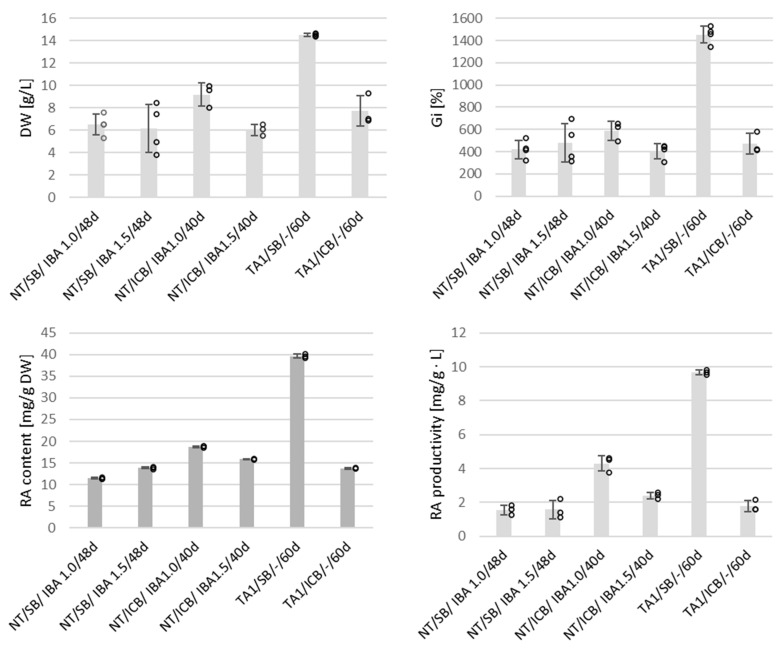
Growth parameters (DW and Gi) of *S. apiana* root cultures, along with rosmarinic acid (RA) content [mg/g DW] and productivity [mg/L∙day], cultivated in SH_0_ medium (TA1 clone of hairy roots) and SH medium with 1.0 or 1.5 mg/L IBA (non-transformed, adventitious roots—NT). Cultures were maintained in ICB and SB systems, during 40–60 days of cultivation. Data are presented as the mean of at least three replicate experiments, with bars indicating standard errors and dots representing individual replicates.

**Table 2 ijms-26-03138-t002:** Rosmarinic acid and cryptotanshinone contents [mg/g DW] of roots of the intact plant and in vitro root cultures of *S. apiana* (*n* = 3).

Compound	Roots of the Intact Plant	Adventitious Root Culture	Transformed Root Culture				
			TA1	TA2	TA3	TL1	TL2
Rosmarinic acid	ND	17.22 ± 0.33	12.76 ± 0.09	11.01 ± 0.28	9.83 ± 0.19	15.32 ± 0.39	12.97 ± 0.06
Cryptotanshinone	3.71 ± 0.20	ND	ND	ND	ND	ND	ND

ND—not detected.

**Table 3 ijms-26-03138-t003:** Experimental design for *Salvia apiana* root cultures in the studied in vitro systems. All cultures were grown in liquid SH medium supplemented with 3.0% (*w*/*v*) sucrose.

Experiment Designation	Root Line ^a^	Culture Type	Lighting Conditions	PGRs (mg/L)	Growth Period (days)	Harvesting Interval (days)	No of Replicates Per Treatment	Phytochemical Analysis (day) ^b^
EX0	NT	agitated	darkness	IBA (1.0)	21	-	-	21
	TA1	agitated ^c^	darkness	-	21	-	-	21
	TA2	agitated ^c^	darkness	-	21	-	-	21
	TA3	agitated ^c^	darkness	-	21	-	-	21
	TL1	agitated ^c^	darkness	-	21	-	-	21
	TL2	agitated ^c^	darkness	-	21	-	-	21
EX1	NT	agitated ^c^	darkness	IBA (0.5–5.0 with 0.5 increments)	48	-	6	-
EX2	NT	agitated ^c^	darkness	IBA (1.0)	69	3	3	48
	NT	agitated ^c^	darkness	IBA (1.5)	69	3	3	33
	NT	agitated ^c^	photoperiod	IBA (1.0)	45	3	3	-
EX3	NT	ICB ^d^	darkness	IBA (1.0)	40	-	3	40
	NT	ICB ^d^	darkness	IBA (1.0)	48	-	3	48
	NT	ICB ^d^	darkness	IBA (1.5)	40	-	3	40
	NT	ICB ^d^	darkness	IBA (1.5)	48	-	3	48
	NT	SB ^e^	darkness	IBA (1.0)	40	-	3	40
	NT	SB ^e^	darkness	IBA (1.0)	48	-	3	48
	NT	SB ^e^	darkness	IBA (1.5)	40	-	3	40
	NT	SB ^e^	darkness	IBA (1.5)	48	-	3	48
EX4	TA1	agitated ^c^	darkness	-	60	10	6	50
	TA2	agitated ^c^	darkness	-	60	10	6	-
	TA3	agitated ^c^	darkness	-	60	10	6	-
	TL1	agitated ^c^	darkness	-	60	10	6	-
	TL2	agitated ^c^	darkness	-	60	10	6	-
EX5	TA1	ICB ^d^	darkness	-	60	-	3	60
	TA1	SB ^e^	darkness	-	60	-	3	60

^a^ NT, adventitious roots; TA1, TA2, TA3, hairy roots obtained using A4 strain; TL1, TL2, hairy roots obtained using LBA9402 strain. ^b^ Harvest day of samples subjected to phytochemical analysis. ^c^ 100 mL Erlenmeyer flasks closed with silicone foam stoppers, 25 mL working volume, 1:40 (*m*/*v*) root-to-medium ratio 120 rpm (25.4 mm orbit). ^d^ Immersion-column bioreactor, 600 mL working volume, 1:40 (*m*/*v*) root-to-medium ratio, 0.3 L/min aeration rate (constant aeration). ^e^ Spray bioreactor, 600 mL working volume, 1:40 (*m*/*v*) root-to-medium ratio, 0.3 L/min aeration rate (5 min every 1.5 h), medium dispersal rate 100 mL/min (5 min every 1.5 h).

## Data Availability

The datasets presented in this article are not readily available because the data are part of an ongoing study and have not been archived yet. Requests to access the datasets should be directed to corresponding author: agata.krol@gumed.edu.pl.

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
