# Peer review of "Adventitious and Hairy Root Cultures of Salvia apiana as a Source of Rosmarinic Acid"

_ijms, 2025, doi:10.3390/ijms26073138_

Round 1
Reviewer 1 Report
Comments and Suggestions for Authors
For authors
Dear authors, please find in next my recommendations for the manuscript "Adventitious and Hairy Root Cultures of Salvia apiana as a Source of Rosmarinic Acid"
Introduction section:
The introduction section of the manuscript provides an overview of the significance of the Salvia genus in medicine, highlighting its rich repository of secondary metabolites with relevant biological activities. The author introduces the S. apiana Jeps. (subg. Audibertia), an endemic species for SW part of USA and NW part of Mexico.
I recommend that the authors reorganise the hypothesis and research objectives (L 66-82) to be more fluent for the reader. Information about “in vitro root cultures of Salvia spp” – “metabolites profile”, “phytochemical profile of transformed roots….” – see L 68-73, in my opinion, should be presented and detailed in an earlier paragraph. Also, “available data on the chemical composition of S. apiana” (L74) could be given to readers as supplementary material or as a brief description before the hypothesis and research question paragraphs.
Results and discussion section:
In this section, the manuscript details the establishment of both adventitious and hairy root cultures of Salvia apiana, underscoring successful in vitro propagation techniques implemented through meticulous methodology. The results illustrate quantitative data on biomass production and the accumulation of key secondary metabolites, particularly rosmarinic acid, under various culture conditions. The discussion part critically analyses the performance metrics, noting that while the cultures achieved significant yields of rosmarinic acid, the anticipated production of diterpenoids was not observed. The authors compare these results with existing literature on other Salvia species, offering potential explanations such as differences in growth conditions, culture system efficiencies, and genetic variability inherent to in vitro cultures. Moreover, the section identifies specific experimental limitations and addresses areas where further optimisation, such as improved bioreactor parameter control and elicitation strategies, could bolster metabolite synthesis.
However, in my opinion, some parts of the results and discussions section could be improved through:
1.) To provide a more direct comparison and discussion of results with similar studies on other Salvia species - e.g. metabolite production, growth characteristics, etc. More precise comparisons of experimental data with others available in the literature will ensure public understanding of the specific phytochemistry of S. apiana
2.) The manuscript should better explain why diterpenoids were not detected in all crops, considering factors such as culture conditions, gene expression or growth stages (please explain more accurately L251-255)
3.) The manuscript should better emphasise the results obtained and their importance. In this respect, the authors should also consider the sustainable production of bioactive compounds (e.g. rosmarinic acid) and their applicability in various sectors. They should also clearly indicate the study's limitations, such as the absence of diterpenoid production or possible bioreactor inefficiencies and suggest experimental modifications or future studies to address this limitation. Last, the author should propose specific future research directions to overcome the challenges related to the manuscript's main area of interest.
Methods section:
Please provide more details about the HPLC-DAD-ESI-MS analysis of non-volatile phenolics, such as recovery, accuracy, linearity, etc. (They could be introduced as a Table in Supplementary material)
Considering the above-mentioned, I propose this manuscript for “Reconsider after major revisions (substantial revisions to text or experimental methods needed)”
Comments on the Quality of English LanguageAs a non-native speaker, I am not able to judge this correctly. However, I didn't have difficulties to read and understand the attached manuscript
Reviewer 2 Report
Comments and Suggestions for Authors
The manuscript reports a study on content of secondary metabolites in adventitious and hairy root cultures of Salvia apiana (white sage). It is a innovative contribution of good scientific and technical value. Material and methods and results are well described.
I have to suggest some improvement on the discussion of the results:
Lines 110 – 116. In the present study no hairy roots were observed after infection with the ATCC15834 strain, thus, I think that the consistence "with previous studies demonstrating that sage plants generally show high susceptibility to transformation by R. rhizogenes, especially strains ATCC15834…. “should be reconsidered.
Lines 157-166. … HPLC analysis revealed the presence of 6 diterpenoids in the extracts from roots of the intact plant and the main metabolite of this group was cryptotanshinone which was not found in root cultures either transformed or not with R. rhizogenes. On the other hand, the extract from roots of field-grown plant was found not containing phenolic acids, including rosmarinic acid, despite the fact that this compound had previously been identified in the underground parts of white sage, however, rosmarinic acid was found in root cultures either transformed or not. Concerning these results, I would improve the discussion trying to explain why the in vitro conditions in either transformed or not root culture lines can induce these modifications of contents in these metabolites.
Round 2
Reviewer 1 Report
Comments and Suggestions for Authors
Dear Authors,
Thank you for the seriousness with which you have considered and dealt with the mentioned recommendations. I believe that the requested improvements have been well realised in all main sections of the manuscript (introduction, results and discussion, methods). After reading the current form of the manuscript, I did not find any relevant or significant parts that should be proposed for further analysis. Considering these issues, I recommend to the Editor that the current form of this manuscript could be considered for publication in the IJMS journal, as it fulfils the rigorous scientific quality requirements of the journal.
Best regards
Reviewer 2 Report
Comments and Suggestions for Authors
The manuscript was accurately revised considering all the suggestions and I think that it is suitable for publication